# Scapular Dyskinesis in Elite Boxers with Neck Disability and Shoulder Malfunction

**DOI:** 10.3390/medicina57121347

**Published:** 2021-12-09

**Authors:** Jae Woo Jung, Young Kyun Kim

**Affiliations:** Graduate School of Sports Medicine, CHA University, Seongnam 13496, Korea; jaewoojung0923@gmail.com

**Keywords:** boxing, scapular dyskinesis, shoulder range of motion, neck pain

## Abstract

*Background and Objectives:* Neck and shoulder injuries commonly occur during boxing, and scapular dyskinesis is related to those injuries. This study investigated scapular dyskinesis with neck disability and shoulder malfunction in elite boxers. *Materials and Methods:* Seventy-two elite boxers participated in this study. Scapular dyskinesis was evaluated as normal, subtle, and obvious. Neck disability index (NDI), shoulder internal (IR), and external (ER) range of motion (ROM), isometric strength of IR and ER, and pectoralis minor length were measured and compared with the severity of scapular dyskinesis. *Results:* Thirty-eight boxers (52.7%) showed scapular dyskinesis. NDI score was significantly different (normal = 3.89 ± 3.08, obvious = 7.36 ± 4.95, *p* = 0.025). Isometric IR strength was significantly different (normal = 10.48 ± 2.86, obvious = 8.46 ± 1.74, *p* = 0.01). The length of the pectoralis minor was significantly different (normal = 10.17 ± 0.67, subtle = 9.87 ± 0.79, obvious = 9.47 ± 0.85; *p* = 0.001), and the dominant and non-dominant arm IR ROM was significantly different (dominant = 57.43 ± 11.98, non-dominant = 64.62 ± 10.3, *p* = 0.001). *Conclusions:* The prevalence of scapular dyskinesis is high among elite boxers. Boxers with scapular dyskinesis presented shoulder malfunction as well as neck disability. Further investigation is necessary to examine the relationship between scapular dyskinesis and neck disability in boxers.

## 1. Introduction

Although boxing could lead to traumatic injuries, boxers sustain less frequent injuries compared to athletes of other contact sports [1,2,3]. However, boxers are allowed punches to the head, which could cause acute and chronic injuries to the head, neck, and shoulders [1]. Headgears, shorter rounds, mouth guards, and increased glove size reduce the risk of boxing injuries [1,4]. However, rates of head, neck (34.2%), and shoulder injuries (9.7%) remain high [1]. Punching accounts for 36.8% of all boxing injuries, and head and neck injuries are related to chronic neurological injuries [1,2]. Shoulder subluxation and dislocation injuries are reported to be a critical physical limitation to boxers [5]. Boxers miss 14.2–20 training days on average owing to shoulder injuries [5,6]. Indeed, frequent boxing injuries could lead to decreased strength in boxers and limit the range of motion (ROM) in their shoulder, resulting in scapular dyskinesis [7].

Scapular dyskinesis is defined as abnormal static position and/or dynamic movement of the scapula [8]. McClure et al. categorized scapular dyskinesis into three grades, namely normal, subtle, and obvious abnormality to differentiate the severity of scapular dyskinesis [9]. In 2014, Clarsen et al. reported that obvious scapular dyskinesis is associated with reduced total shoulder rotational ROM, reduced external rotation strength of the shoulder, and increased the risk of shoulder injury among elite male handball players [10]. The risk of shoulder injury increased with subtle (OR, 3.48; 95% CI, 0.83–14.5; *p* = 0.09) and obvious abnormality (OR, 8.41; 95% CI, 1.47–48.1; *p* = 0.02) of scapular dyskinesis among handball players. Kawasaki et al. in 2012, reported that scapular dyskinesis was associated with discomfort in the shoulder (OR, 4.4), and an asymptomatic shoulder with scapular dyskinesis revealed higher rate of discomfort within the shoulder during playing season (OR 3.6) among rugby players [11]. In boxers, the prevalence of scapular dyskinesis is reportedly 2.73 times higher than among non-boxers [7]. Previous studies show that scapular dyskinesis is related to shoulder dysfunction in athletes. Therefore, understanding the characteristics of scapular dyskinesis is important to reduce the risk of shoulder injuries among boxers.

Previously, researchers have reported factors that may lead to scapular dyskinesis. Factors identified with scapular dyskinesis include thoracic kyphosis and increased cervical lordosis, alteration in scapular muscle function, decreased flexibility of muscles around the scapula [12], glenohumeral internal rotation deficit (GIRD) [13], decreased subacromial space [14,15], shoulder impingement [16,17], and decreased shoulder strength [18,19,20]. Baseball pitchers with insufficient external rotation ROM have a 2.2-fold increased risk of shoulder injury, and a 4-fold increased chance of undergoing surgery [21]. Additionally, increased internal rotation reduces the risk of injury by 22–63%, and injured athletes show decreased total arc and internal rotation of the dominant arm, as compared to non-injured athletes [22]. Moreover, decreased external rotation strength also increases the risk of injury by 1.4 times [10]. Therefore, shoulder malfunction is related to scapular dyskinesis and it is important for boxers to recover and prevent shoulder injuries. Neck injury is also related to scapular dyskinesis since scapular position and motion could be altered owing to neck pain [23,24,25,26]. Head, neck, and shoulder injuries are common in boxing [1]. However, no study has investigated the effect of neck disability on scapular dyskinesis in boxing.

Scapular dyskinesis is common in overhead athletes [13]. Although boxing is not an overhead sport, repetitive punching and being repeatedly punched in the face could cause microtrauma to the shoulders and neck. Repeated microtrauma could cause anterior capsule laxity and posterior capsule tightness due to continuous eccentric loading that could lead to scapular dyskinesis [27]. Lenetsky et al. in 2015, reported that boxers presented with weaker shoulder rotation strength, decreased shoulder rotational ROM, and higher prevalence of scapular dyskinesis as compared to non-boxers [7]. However, only 18 boxers had participated in that study, and a larger sample size is necessary to identify symptoms related to scapular dyskinesis, including neck disability. Therefore, the purpose of this study was to identify scapular dyskinesis-related shoulder function and neck disability in elite boxers.

## 2. Materials and Methods

### 2.1. Participants and Study Design

This was a cross-sectional, single blinded study. We listed all elite boxing teams in the Republic of Korea and reached out to them using posters about the study to recruit participants. There are 263 registered elite male boxers at Korea Boxing Federation in the Republic of Korea; of these, 72 (144 shoulders) boxers volunteered to participate in this study. The inclusion criteria were as follows: elite boxers with four or more years of boxing experience. The exclusion criteria were amateur boxers, head or neck injury in the past three months or upper body surgery in the past one year. We required a sample size of 63 for an effect size 0.97, significance level of 0.05, and a power of 0.95 after calculating the sample size with G Power (University of Kiel, Kiel, Germany). We, however, recruited a larger sample size of boxers. An informed consent form was given prior to data collection. The CHA University’s ethical review board approved this study (1044308-202010-HR-046-02).

### 2.2. Protocol

The demographic data of all participants, including age, height, weight, and boxing experience (years) were obtained. Next, the presence of scapular dyskinesis was evaluated using the scapular dyskinesis test (SDT). According to the SDT results, participants were divided into three groups (1-normal, 2-subtle, and 3-obvious scapular dyskinesis). The grouping results were blinded from other measurements to prevent bias from other investigators. After SDT, the participants were asked to answer the neck disability index (NDI) questionnaire, following which, information was obtained on the internal and external shoulder rotational ROM, the length of pectoralis minor, and the isometric strength of internal and external shoulder rotation measurements. The independent variable was the grade of SDT (normal, subtle, and obvious). Dependent variables were other followed measurements. We compared the average of NDI scores, shoulder internal and external ROM and strength, and the length of the pectoralis minor to the severity of scapular dyskinesis to analyze the differences.

### 2.3. The SDT

The participants were asked to remove their shirts and hold the dumbbells in both their hands. The weight of the dumbbells was decided based on the weight of the subjects. Participants who weighed less than 68.1 kg were handed 1.4 kg (3 lb) dumbbells, and those who weighed more than 68.1 kg were given dumbbells that weighed 2.3 kg (5 lb). While standing upright, the participants were asked to flex their shoulders with their elbows straight in the thumbs up position until 180 degrees of flexion and returned to the starting position for 5 s, and this was repeated five times [9,28]. The investigator stood 1 m behind the participant to measure the results. The results were categorized as follows: 1-normal scapular rhythm, 2-subtle abnormal pattern, and 3-obvious abnormal pattern according to McClure (2009) (Figure 1) [9,29]. The investigator for SDT was a certified athletic trainer with 13 years of experience in the field and its research. The results of SDT were blinded from other investigators and the reliability of SDT was high (ICC = 0.86) [30].

### 2.4. The NDI

The NDI includes 10 questions measuring neck pain-related disability [31]. It is the most widely applied questionnaire for neck pain, with high reliability (ICC = 0.89) [32,33]. The NDI score is between 0 and 50, and a higher score indicates higher disability due to neck pain [33,34]. All participants completed the Korean version of the NDI to measure neck disability; the reliability of the Korean NDI is very high (ICC = 0.927) [34].

### 2.5. Shoulder Internal (IR) and External (ER) ROM

The participant’s dominant and non-dominant shoulder IR and ER ROM were measured in the supine position. The dominant arm was decided to be the rear arm in a boxing stance [7]. The shoulder was abducted at 90 degrees and the elbow was flexed at 90 degrees. The digital inclinometer (EX-POWER, Ansan, Korea) was placed in the mid-point of the forearm. The first investigator stabilized the shoulder against the table, and the second investigator held the participant’s distal forearm to produce maximum passive rotation [35]. The maximum IR and ER were measured when the scapula began to move [10]. Two practices were initiated, then two repeated measures were recorded, and their average was used for IR and ER ROM measurements. The total rotational ROM was calculated by the sum of IR and ER ROM. The reliability of IR and ER measurements was high (ICC = 0.85–0.99) [36].

### 2.6. Shoulder Isometric Strength Test

Micro FET 2 hand-held dynamometer (HHD) (Hoggan Health Industries INC., Salt Lake City, UT, USA) was used to measure shoulder isometric strength of IR and ER. The participant was in supine position with the shoulder in neutral position and the elbow flexed at 90 degrees. The investigator stabilized the upper arm, pushing down to the table. The HHD was located at 5 cm, proximal to the wrist on dorsal (ER) or ventral (IR) strength measure. The participant was asked to gradually increase the resistance to maximum. Two practice sessions followed by two final measurements were applied to measure the isometric strength of ER and IR [37]. The reliability of the IR and ER isometric strength measurements was high (ICC = 0.85–0.99) [36]. To normalize muscle strength with the body size, isometric strength was divided by the participant’s weight to the 0.67th power. We used the formula recommended in the strength test [38].

### 2.7. Pectoralis Minor Length

The participant was in supine position and was allowed to rest to relax for one minute prior to data collection [39]. The medial-inferior angle of the coracoid process and the lateral sternocostal junction of the inferior fourth rib were marked. We used a caliper to measure the distance between the two marks. To normalize the length of the pectoralis minor, the length was divided by the participant’s height and multiplied by 100. The reliability of the length of the pectoralis minor was high (ICC = 0.82–0.87) [40]. We measured the length of the pectoralis minor since the tightness of the pectoralis minor is related to scapular dyskinesis [13,41].

### 2.8. Statistical Analysis

SPSS 22.0 (SPSS Inc., Chicago, IL, USA) was used for statistical analyses. To compare the average of neck disability, shoulder rotational ROM and isometric strength, and the length of the pectoralis minor by the severity of scapular dyskinesis, a one-way ANOVA was used to analyze the data. The Shapiro–Wilk test was used and the data were normally distributed. Bonferroni and LSD post-hoc tests were used to compare each group when the results were significantly different. A paired *t*-test was used to compare IR and ER ROM of dominant and non-dominant arms. A normal distribution test was generated since the total sample size was 72 volunteers (with 144 shoulders). The level of significance was set at *p* < 0.05.

## 3. Results

A total of 72 elite Korean boxers participated in this study. Thirty-eight boxers (52.7%) showed scapular dyskinesis in their shoulders. Thirty-four boxers (47.22%) were normal, thirty-three (45.8%) were subtle, and five (6.94%) showed obvious scapular dyskinesis in the dominant arm. Thirty-five boxers (48.51%) were normal, twenty-six (36.11%) were subtle, and eleven (15.27%) showed obvious scapular dyskinesis in the non-dominant arm (Table 1).

The associations between NDI, IR/ER ROM and strength, pectoralis minor length with scapular dyskinesis are presented in Table 2. The NDI score of the non-dominant arm with scapular dyskinesis was significantly different between the normal and obvious group (normal = 3.89 ± 3.08; obvious = 7.36 ± 4.95; *p* = 0.025). However, the NDI score of the dominant arm with scapular dyskinesis was not significantly different between the normal, subtle, and obvious groups (normal = 5.15 ± 4.27; subtle = 4.55 ± 3.49; obvious = 7.8 ± 5.07; *p* = 0.238). We found no significant difference in IR ROM (normal = 60.36° ± 11.51; subtle = 62.16° ± 11.96; obvious = 59.67° ± 11.98; *p* = 0.61) and ER ROM (normal = 97.15° ± 15.24; subtle = 94.36° ± 16.47; obvious = 94.64° ± 16.15; *p* = 0.58). Although isometric IR strength was significantly different (normal = 10.48 ± 2.86; obvious = 8.46 ± 1.74; *p* = 0.01), isometric ER strength was not significantly different (normal = 10.03 ± 2.32, subtle = 9.87 ± 0.79, obvious = 9.47 ± 0.85, *p* = 0.39). The length of the pectoralis minor was significantly different between the normal and subtle groups (normal = 10.17 ± 0.67, subtle = 9.87 ± 0.79, obvious = 9.47 ± 0.85, *p* = 0.023), and normal and obvious (*p* = 0.001) scapular dyskinesis. There was a tendency of difference between the subtle and obvious groups (*p* = 0.054) and the length of the pectoralis minor.

There was a significant difference between the dominant and non-dominant arm’s IR ROM (dominant = 57.43 ± 11.98, non-dominant = 64.62 ± 10.3, *p* = 0.001) (Table 3). There was no significant difference between the dominant and non-dominant arm’s ER ROM (dominant = 95.8 ± 15.83, non-dominant = 95.65 ± 15.87, *p* = 0.915)

## 4. Discussion

The purpose of this study was to investigate the incidence of scapular dyskinesis in boxers with neck disability and shoulder malfunction. We identified a high incidence of scapular dyskinesis (52.7%, N = 72) in boxers with neck disability and shoulder malfunction including decreased ROM, strength, and muscle length. The prevalence of scapular dyskinesis in overhead and non-overhead athletes was 61% and 33%, respectively [42]. Overhead athletes showed greater prevalence of scapular dyskinesis owing to repetitive overhead swings that can cause increased stress and damage to the shoulder [42,43,44]. Repeated high velocity movement, such as punching, could cause microtrauma and capsule laxity, leading to scapular dyskinesis [13,45]. In addition, we found 11 cases of obvious scapular dyskinesis in the non-dominant arm, as compared to the dominant arm (5 obvious) owing to more frequently launched punching to the non-dominant arm [46]. Therefore, high scapular dyskinesis rate in boxers could be due to repetitive punching.

We found that boxers with scapular dyskinesis in the non-dominant shoulder showed significantly higher NDI scores (normal = 3.89 ± 3.08, obvious = 7.36 ± 4.95, *p* = 0.025). Although the NDI scores depicting obvious scapular dyskinesis presented mild disability in the neck, boxers with normal scapular dyskinesis showed no disability in their neck (NDI 0–4 = no disability, 5–14 = mild disability) [32]. Scapula provides mobility and stability to the shoulder and neck, and acts as a bridge [41]. Neck pain could affect the muscles attached between the cervical spine and scapula, causing altered scapular motion [41,47]. According to Yildiz (2019), patients with neck pain presented with altered scapular motion compared to healthy individuals (48). Moreover, scapular dyskinesis in the neck pain group showed significant lower middle trapezius activity with scapular retraction compared to the healthy group with scapular dyskinesis (*p* = 0.029) [47]. Ha et al. (2011) and Van Dillen et al. (2007) reported that corrected scapula position significantly reduced neck pain in patients [24,48,49]. Therefore, our results support the relationship between neck disability and scapular dyskinesis. According to our results, cases of scapular dyskinesis in the dominant shoulder were not significantly different. However, punches with the non-dominant shoulder are more frequently launched in boxing as compared to the dominant shoulder (*p* = 0.013) [50]. Andres et al. reported that repeated shoulder motion increases the rate of scapular dyskinesis (*p* = 0.002, *p* = 0.033) [51]. Therefore, our results are consistent with those of previous studies and report that boxers with obvious scapular dyskinesis were more prone to having neck disability as compared to those with normal scapular dyskinesis.

IR ROM and ER ROM were not significantly different depending on scapular dyskinesis in boxers. However, IR ROM between the dominant and non-dominant arm was significantly different (dominant = 57.43 ± 11.98, non-dominant = 64.62 ± 10.3, *p* = 0.001). Several previous studies have reported that overhead athletes presented with decreased IR ROM [52,53,54,55]. Borsa et al. [53] reported decreased IR ROM (dominant = 59.7 ± 7.0, non-dominant = 68.2 ± 8.6, *p* = 0.008) in baseball pitchers. In this regard, our results are similar to those from previous studies (dominant = 57.43 ± 11.98, non-dominant = 64.62 ± 10.3, *p* = 0.001). Lenetsky et al. also reported decreased IR ROM in the dominant arm as compared to the non-dominant arm in boxers [7]. Decreased IR ROM is seen in the dominant shoulder of overhead athletes owing to repetitive overhead swing motion causing microtrauma to the shoulder capsule [27]. Although punching in boxing and overhead throwing are different, there are similarities such as high velocity repetitive movement of shoulder [7,56], and it causes anterior capsule laxity and increases the risk of shoulder injury [45]. Repeated eccentric loading is also known to increase tightness in the shoulder’s posterior capsule, and these injuries cause limited IR ROM in shoulders of overhead throwers and boxers [7,27]. Therefore, based on our results, boxers also present with shoulders that have limited IR ROM, which is similar to the shoulders of overhead throwers, owing to similar mechanism [7].

We also found decreased shoulder IR isometric strength in cases with obvious scapular dyskinesis (normal = 10.48 ± 2.86, obvious = 8.46 ± 1.74, *p* = 0.01). Decreased rotator cuff strength is found in cases of scapular dyskinesis [8]. Multiple factors are known to cause scapular dyskinesis, including decreased rotator cuff strength [13]. According to Kibler et al. (2006), rotator cuff strength could be increased or decreased depending on the scapular position [19]. Fatigued rotator cuff muscles could alter scapular position and this could lead to shoulder impingement, diminishing the rotator cuff function [17]. However, several studies reported no significant difference in IR isometric strength in baseball pitchers (dyskinesis = 131.3N ± 41.7, normal 139.4N ± 33.6, *p* = 0.911) and the healthy population (*p* = 0.34) with and without scapular dyskinesis [57,58]. According to Smith et al. (2006), protracted scapular position decreased isometric internal rotation strength [59]. Protracted scapula may have affected isometric internal rotation strength measurement. Further investigation is necessary to reveal the relationship between scapular dyskinesis and rotator cuff strength.

Shortened pectoralis minor could alter normal scapular motion and decrease the subacromial space [60]. We found significantly decreased length of the pectoralis minor with increasing severity of scapular dyskinesis (normal = 10.17cm ± 0.67; subtle = 9.87cm ± 0.79; obvious = 9.47cm ± 0.85). Tightness of the pectoralis minor is related to scapular dyskinesis [13,41,61]. A previous study revealed that participants with scapular dyskinesis showed a significantly decreased length of pectoralis minor as compared to normal participants (scapular dyskinesis = 7.49 ± 0.38, normal = 8.58 ± 0.75, *p* = 0.001) [62]. Our results showed that the length of the pectoralis minor decreases as scapular dyskinesis becomes worse. Therefore, lengthening the pectoralis minor could help in treating the shortness in boxers. However, rounded-shoulder might be the nature of boxing. Further research is required to investigate the effects of pectoralis minor tightness with scapular dyskinesis in boxers.

We recruited a large number of participants (N = 72) to investigate scapular dyskinesis-related malfunction in the neck and shoulder in boxers. However, there is no known gold standard test for scapular dyskinesis, and this could be the limitation of this study. A recent reliability study of SDT with the three grades (normal, subtle, and obvious) reported high inter-reliability (ICC = 0.86) (30). Kibler et al. [63] suggested an SDT with four categories; however, we decided to use the three-grade test due to its higher reliability. Our results may have therefore been different from Kibler’s test comprising four categories. Additionally, we did not investigate the history of injury and training volume and type. Further investigation is required to detect the relationship between scapular dyskinesis and other possible variables in boxing.

## 5. Conclusions

Boxers showed a 52.7% rate of incidence of scapular dyskinesis. The prevalence rate was lower than that seen with overhead athletes, but higher than that seen with non-overhead athletes. Boxers with scapular dyskinesis showed increased neck disability and decreased internal rotation ROM and strength, along with reduced pectoralis minor length. Scapular dyskinesis is identified in many shoulder injuries, and neck disability is also an important factor for the occurrence of scapular dyskinesis in boxers. Therefore, we recommend monitoring scapular dyskinesis in boxers to treat and prevent shoulder and neck injury. Further investigation is required to examine the relationship between scapular dyskinesis and neck and shoulder injury in boxers.

## Figures and Tables

**Figure 1 medicina-57-01347-f001:**
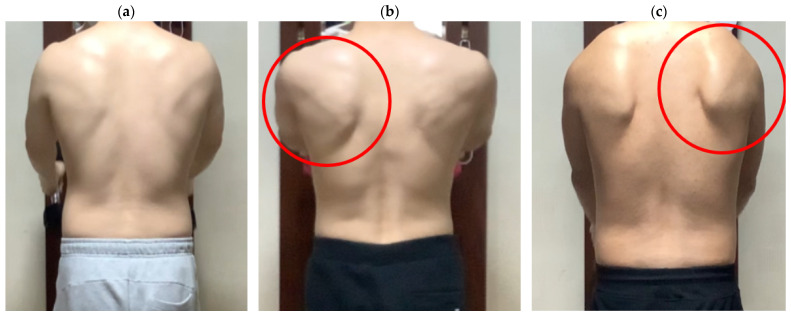
Scapular dyskinesis test rating. (**a**) Normal; (**b**) Subtle (in red circle); (**c**) Obvious (in red circle).

**Table 1 medicina-57-01347-t001:** Prevalence of scapular dyskinesis.

Scapular Dyskinesis	Normal	Subtle	Obvious
Dominant arm (N = 72)	34 (47.22%)	33 (45.83%)	5 (6.94%)
Non-dominant arm (N = 72)	35 (48.51%)	26 (36.11%)	11 (15.27%)

**Table 2 medicina-57-01347-t002:** NDI, IR/ER ROM and strength, pectoralis minor length with scapular dyskinesis.

Scapular Dyskinesis	Normal ^a^ (95% CI)	Subtle ^b^ (95% CI)	Obvious ^c^ (95% CI)	F	*p*	Bonferroni	*p*	LSD
D NDI (Score) (N = 72)	5.15 ± 4.27 (3.66–6.44)	4.55 ± 3.49 (3.31–5.78)	7.8 ± 5.07 (1.51–14.09)	1.466	0.238			
ND NDI (Score) (N = 72)	3.89 ± 3.08 (2.83–4.94)	5.65 ± 4.29 (3.92–7.39)	7.36 ± 4.95 (4.04–10.69)	3.894	0.025 *	a < c		
IR ROM (degree) (N = 144)	60.36 ± 11.51 (57.60–63.12)	62.16 ± 11.96 (59.04–65.27)	59.67 ± 11.98 (53.28–66.05)	0.495	0.611			
ER ROM (degree) (N = 144)	97.15 ± 15.24 (93.49–100.81)	94.36 ± 16.47 (90.07–98.66)	94.64 ± 16.15 (86.03–103.24)	0.535	0.587			
ISO IR (Nm) (N = 144)	10.48 ± 2.86 (9.79–11.17)	9.63 ± 2.47 (8.98–10.27)	8.46 ± 1.74 (7.54–9.39)	4.462	0.013 *	a < c		
ISO ER (Nm) (N = 144)	10.03 ± 2.32 (9.47–10.58)	9.44 ± 2.54 (8.78–10.10)	9.63 ± 2.52 (8.29–10.97)	0.935	0.395			
Pec m length (cm) (N = 144)	10.17 ± 0.67 (10.01–10.33)	9.87 ± 0.79 (9.67–10.07)	9.47 ± 0.85 (9.01–9.92)	6.794	0.023 * 0.001 ** 0.054	a < b a < c b < c	0.023 * 0.001 ** 0.054	a < b a < c b < c

D, dominant arm; ND, non-dominant arm; NDI, neck disability index; IR, internal rotation; ER, external rotation; ROM, range of motion; ISO, isometric; Pecm, pectoralis minor. * *p* < 0.05, ** *p* < 0.001.; a = normal, b = subtle, c = obvious.

**Table 3 medicina-57-01347-t003:** IR and ER ROM with dominant and non-dominant arm.

	Arm	Degrees	t (*p*)
IR ROM (N = 72)	D	57.43 ± 11.98	−5.045 (0.001) ***
	ND	64.62 ± 10.3	
ER ROM (N = 72)	D	95.8 ± 15.83	0.107 (0.915)
	ND	95.65 ± 15.87	

D, dominant arm; ND, non-dominant arm; IR, internal rotation; ER, external rotation; ROM, range of motion. *** *p* < 0.001.

## Data Availability

Participants were enrolled from CHA University Sports Medicine Graduate School Athletic Training Research Laboratory. The data presented in our study are available on request from the first and corresponding author.

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
