# Peer review of "Scapular Dyskinesis in Elite Boxers with Neck Disability and Shoulder Malfunction"

_medicina, 2021, doi:10.3390/medicina57121347_

Round 1
Reviewer 1 Report
Dear authors:
Congratulations for your job. It seems to me of very good methodological quality. I only have one question about it.
- In Material and Methods you write: (Lines 84-85)
“The inclusion criteria were as follows: elite boxers with > 4 years of boxing experience…”
Why was this moment of time chosen? Is there evidence to justify it?
Thank you very much.
Author Response
Thank you for your review.
We have made some corrections in the attached comments

Reviewer 2 Report
This topic is interesting, but some concerns are needed to address as follow,
1. Is scapular dyskinesis really a critical issue on boxers? Since round shoulder, and thoracic kyphosis for boxers is a good protective posture to be punched by opponent. For practical application, based on the current data, should we suggest coach and boxers to correct this "look-like abnormal" posture or scapular pattern? Please discuss.
2. How to perform the scoring ONLY by visual on-the-field assessment : 1-normal scapular rhythm, 2-subtle abnormal pattern, and 3-obvious abnormal. Who assessed by a experienced professionals? In addition, please provide the represenetive photos for each condition to make reader recognize easier
3. Please clarify the order of tests?
4. The ICC of each test MUST be provided for this study rather than previous studies.
5. How about the relationship or correlation between those outcomes? Please clarify.
6. Did author check the neck range of motion and/or muscle strength? and upper trapezius stiffness condition? Since it is difficult to say " neck disability is related to scapular dyskinesis" based on current data.
7. Abstract "Identifying scapular dyskinesis is recommended to treat and prevent neck and shoulder injuries among elite boxers" This is too oversuggested, please rewrite.
Author Response
Thank you for your review
We have made some correction with attached file.

Reviewer 3 Report
The following is the summary of this manuscript:
Background and Oobjectives: Neck and shoulder injuries commonly occur during in boxinging, and scapular dyskinesis is related to those injuries. This study investigated scapular dyskinesis with neck disability and shoulder malfunction in elite boxers. Materials and Methods: Seventy-two elite boxers were participated in this study. Scapular dyskinesis was evaluated as normal, subtle, and obvious. Neck disability index (NDI), shoulder internal (IR), and external (ER) range of motion (ROM), isometric strength of IR and ER, and pectoralis minor length were measured to and compared with with the severity of scapular dyiskinesis. Results: Thirty-eight boxers (52.7%) showed scapular dyskinesis. NDI score was significantly different (normal=3.89±3.08, obvious=7.36±4.95, p=0.025). Isometric IR strength was significantly different (normal=10.48±2.86, obvious=8.46±1.74, p=.01).. The length of the pPectoralis minor length was significantly different (normal=10.17±0.67, subtle=9.87±0.79, obvious=9.47±0.85; p=0.001), and the dominant and non-dominant arm IR ROM was significantly different (dominant=57.43±11.98, non-dominant=64.62±10.3, p=.001). Conclusions: The prevalence of scapular dyskinesis is high amongin elite boxers. Identifying scapular dyskinesis is recommended to treat and prevent neck and shoulder injuries in among elite boxers
The article is interesting. I have several comments:
First, there are some grammatical errors. For example, in the first sentence of the abstract, “are” between “commonly” and “injury” should be removed.
Second, in line 85, please give a definition regarding “elite”.
Third, Table 1 has been inserted in a wrong site.
Fourth, in Table 2, I would suggest to add N.S.D in the column of Bonferroni to denote no significant difference among the three groups.
Author Response
Thank you for your comments
We have attached a file with correction.

Round 2
Reviewer 1 Report
Dear authors:
I accept in present form.
Thank you very much for this job.
Reviewer 2 Report
Thank you for the revision.